# ESG performance and green innovation in commercial banks: Evidence from China

Qiliang Wang[1,2], Yingchun Zhang[2,3]*, Yang Li[2,3], Peihao Wang[4]

**1** School of Economic and Management, Qinghai Minzu University, Xining, Qinghai, P. R. China, **2** Qinghai Provincial Key Laboratory of Big Data in Finance and Artificial Intelligence Application Technology, Xining, Qinghai, P. R. China, **3** Business School, Qinghai Institute of Technology, Xining, Qinghai, P. R. China, **4** School of Finance, Shanghai University of Finance and Economics, Shanghai, P. R. China

* zhangych960525@163.com

**Data Availability Statement:** All relevant data are within the manuscript and its Supporting Information files.

## Abstract

Environmental, Social, and Governance (ESG) is closely related to commercial banks' promotion of "dual-carbon" goals and the concept of sustainable development. The impact of ESG performance on commercial banks' support for green innovation remains an issue for in-depth study. This paper studies 36 Chinese commercial banks in China from 2010 to 2021 and finds that the ESG performance of commercial banks can promote green innovation, and this promotion is more obvious when bank remuneration incentives are effective. Meanwhile, this paper verifies the mediating role of the non-performing loan ratio and the Lerner index in it, which provides channel support for ESG to effectively promote green innovation development. This study enriches the existing literature on environmental, social, and governance performance and green innovation in commercial banks and provides new perspectives and directions for future research.

## Introduction

In the post-epidemic era, environmental, social, and governance problems are occurring frequently on a global scale [1], and "Greenhouse Earth" has become a major problem that countries around the world have to face [2]. In this context, how to effectively promote social green innovation has become a key issue to be solved [3]. On this basis, commercial banks, as the core financial institutions of the indirect financial system, assume the important function of a financial reservoir in the economic system, and the green innovation of the society cannot be separated from the full support of the commercial banking system [4]. Therefore, how to protect the environment while ensuring economic development and how to promote green innovation while ensuring high-quality development of commercial banks has become a key concern of scholars and practitioners around the world [5].

The rise of Environmental, Social, and Governance (ESG) information has opened up new possibilities for studying this key concern [6, 7]. Previous scholars have conducted a large number of studies to explore how ESG information acts on the economic system and to seek new ways of development for high-quality economic development and sustainable development of resources. Studies have confirmed that ESG disclosure has a positive impact on listed

**Funding:** (1) the Social Science Foundation of Qinghai Province (Grant No. 22Y013); (2) Xining Federation of Social Science Circles Ideological and Political Research Quality Enhancement Project (Grant No. [2024]5); (3) Graduate Innovation Program of the School of Economic and Management, Qinghai Minzu University (Grant No. 65M2024108). Role of Funder statement: (1) the Social Science Foundation of Qinghai Province (Grant No. 22Y013): The funders had no role in study design, data collection and analysis, decision to publish, or preparation of the manuscript. (2) Xining Federation of Social Science Circles Ideological and Political Research Quality Enhancement Project (Grant No. [2024]5): The funders had no role in study design, data collection and analysis, decision to publish, or preparation of the manuscript. (3) Graduate Innovation Program of the School of Economic and Management, Qinghai Minzu University (Grant No. 65M2024108): The funders had no role in study design, data collection and analysis, decision to publish, or preparation of the manuscript.

**Competing interests:** The authors have declared that no competing interests exist.

companies: specifically, ESG-advantaged firms are more likely to attract investor attention [8], and good ESG performance can significantly reduce the cost of equity capital [9] and the cost of debt [10] and enhance corporate performance and long-term value [11], while also optimizing goodwill impairment of listed firms and improving firms' financial risk and systemic financial risk in the financial system [12]. In further research, corporate innovation is found to be an important mediating mechanism for ESG to enhance corporate performance [13], while corporate green innovation is also an important mediating channel for digital transformation on ESG performance [14, 15]. Therefore, it can be found that how ESG performance affects innovation, especially green innovation, has become a key issue in studying the mechanism of the role of ESG performance on the economic system.

Focusing on ESG and green innovation discussed in this paper, ESG performance is one of the effective criteria for measuring corporate sustainability performance [16], and green innovation is one of the important driving forces for low-carbon transformation and sustainable development of enterprises, and good ESG performance can have an impact on the green innovation performance of enterprises through the mechanisms of cost effect, resource effect and governance effect [17, 18]. For this influence mechanism, previous researchers have broadly formed an effective theory and an ineffective theory from the perspective of risk management, market competitiveness, governance, and incentive mechanism.

Specifically, scholars of the effective theory school believe that corporate ESG performance has a positive impact on corporate green innovation [19, 20]. First, good ESG performance can enhance firms' risk management capabilities and reduce the risk of violating environmental regulations and potential negative impacts, thus releasing more resources to be invested in green innovation [21–23]. Second, ESG advantages can enhance firms' market competitiveness and promote green innovation as parties pay more attention to sustainability issues [24]. Again, good ESG performance usually represents efficient corporate governance, including transparent decision-making processes, effective monitoring mechanisms, and fair distribution of rights and benefits, which in turn facilitates the flow of information within the firm and thus promotes the implementation of green innovation [25–27]. Finally, incorporating ESG objectives into incentive mechanisms, such as performance appraisal indicators for management, can effectively encourage management and employees to participate in and promote green innovation [28]. In addition, scholars have scaled up the above facilitation effects to the national level to study the macro effects [29], or focused on large-scale manufacturing and industrial enterprises to study the micro-mechanisms [30, 31].

In contrast, scholars of the nullist school argue that there is a time-lag effect in the belief that corporate ESG performance has an impact on green innovation, and that green innovation cannot be released in a timely and effective manner by ESG information, and that excessive attention to ESG by listed firms can divert the attention and resources of the firms, and can even hurt green innovation [32, 33]. To summarize, the study of the relationship between ESG information and green innovation has become one of the key topics of in-depth drilling by various research units. However, the existing research literature pays more attention to non-financial institutions [34], and the research results on the relationship between ESG and green innovation of commercial banks are slightly thin, which to a certain extent is not conducive to the development of high-quality paths of the economy from the financial system of indirect finance. Therefore, this paper tries to grasp the special subject of commercial banks to conduct targeted research on the influence mechanism of the above two.

Based on the above research background, this paper will draw on academic research on non-financial institutions to examine the impact of commercial banks' ESG performance on green innovation from the perspectives of risk management, market competitiveness, governance, and incentives. The possible marginal contributions of this paper are as follows: first,

research perspective. Current academic research on banks' innovation drivers is saturated, but little literature has explored the social responsibility perspective. This paper explores the impact of banks' ESG performance on green innovation, providing a new theoretical perspective for academic research in the field of bank innovation. Second, the mechanism. This paper investigates the role of banks' ESG responsibilities, and discusses the impact of banks' ESG performance on green innovation from the perspectives of management compensation (governance and incentive mechanism), non-performing loan ratio (representing risk management), and Lerner's index (reflecting market competitiveness). This theoretical result provides new insights for academic research in this area and fills the previous research gap. Third, policy support. Studying the impact of banks' ESG performance on green innovation helps policymakers to incorporate ESG factors in the regulatory framework, thus guiding the banking industry towards sustainable development.

The remainder of the paper is organized as follows. Section "Theory and hypotheses" presents theory introduction and hypothesis development. Section "Data and methodology" goes over the research methodology. Section "Empirical results and discussions" presents the empirical analysis of this study. Lastly, the section "Conclusion and policy implications" concludes and gives policy implications.

## Theory and hypotheses

### ESG performance and green innovation

ESG performance will make other participants in the economy and society pay attention to the ESG behavior of commercial banks, and take the social responsibility performance of commercial banks as one of the decision-making factors, which creates a new behavioral constraint on commercial banks, and then positively incentivizes the green innovation of commercial banks.

On the one hand, based on the information asymmetry theory, green innovation activities require a long return cycle and the return benefit is not clear, and the problems of adverse selection and moral hazard are particularly serious [35]; therefore, if there are high information barriers inside and outside of the commercial bank, the negative impact on the green innovation activities will be huge [36]. And when commercial banks have better ESG rating performance, externally, external investors can intuitively feel the contribution of commercial banks to society in the field of green environmental protection through ESG ratings [37], which can better promote the formation of a tripartite partnership between the enterprise, the market, and the government, and push commercial banks to further support green R&D investment [38]; from an internal perspective, commercial banks can judge the necessity of green innovation project investment through ESG rating information, reduce the threshold of brown industry's investment in green innovation, and effectively alleviate the innovation dilemma of information asymmetry [39]. In summary, the ESG performance of commercial banks can alleviate the problem of internal and external information barriers, and by improving the information environment, it can then promote the efficiency of green innovation in commercial banks.

On the other hand, based on the theory of limited attention, economic individuals have limited cognitive abilities [40] and need to selectively allocate their attention to different things [41], so investors are more inclined to pay attention to macro-market information while relatively neglecting micro-firm information [42], and identify firms' information partially to third-party organizations. When this cognitive fairness does not exist, economic individuals may not even be able to effectively recognize the information at all [43]. At a time of increasing concern about the social responsibility of commercial banks, the ESG performance of commercial banks given by independent rating agencies can inform a wide range of economic

individuals of the efficiency of a bank's green responsibility. This leads to further capital pooling and supports green innovation of commercial banks, creating positive feedback.

In summary, we suppose that:

**Hypothesis 1.** Commercial banks' ESG performance has a facilitating effect on their green innovation.

## The moderating effect of remuneration incentives

The compensation incentive system implies that if the firm performs well, the management will benefit at the same time, solving the credit risk and principal-agent problem of the management by aligning their interests with those of the firm [44]. Established studies have demonstrated that compensation incentives can effectively promote corporate innovation [45]; thus, commercial banks can positively regulate the support of green innovation from commercial banks' ESG performance by incorporating ESG performance into management compensation assessment indicators [46–48]. Therefore, we suppose that:

**Hypothesis 2.** When commercial banks have higher remuneration incentives, the positive correlation between ESG performance on green innovation is higher.

## The mediating effect of non-performing loan (NPL)

One of the central features of the banking system and the financial crisis is ex-post credit risk in the form of non-performing loans [49, 50], where credit expansion during periods of economic growth is exposed during periods of economic adjustment, which in turn creates non-performing loans [51, 52], exposing banks exposed to significant credit risk, creating structural financial risk to the economy and society [53]. In contrast, banks with good ESG performance care about the interests of their stakeholders and therefore carefully screen borrowers and monitor loan utilization to reduce the risk of default, which in turn reduces the NPL ratio [54–56]. At the same time, the decline in the NPL ratio will enhance the investment confidence of banks, making them more willing to expand the scale of lending, which in turn will synchronize the increase in the volume of loans invested in the green sector [57], eliminating the negative effects of financial barriers on green innovation [58], and facilitating green innovation in the economy and society. Therefore, we suppose that:

**Hypothesis 3.** NPL ratios play a mediating role in commercial banks' ESG performance affecting green innovation.

## The mediating effect of the Lerner index

The Lerner Index is used to measure the degree of market competition of firms in an industry [59, 60]. Established studies have found that commercial banks' top-performing ESG indicators can attract investors' attention and capital inflows, which can drive up the share price, which in turn increases the bank's Lerner Index and enhances its competitive advantage [61–63]. Meanwhile, the Lerner index has a significant contributing role in promoting green innovation. Monopolistic commercial banks with high Lerner indexes usually have more resources and capital [64], and can provide large and targeted green financial products to meet the financial needs of green innovation [65, 66], enabling monopolistic banks to better support and invest in green innovation projects [67], can focus more opportunities on sustainable development and green economy, and provide a solid economic foundation for promoting green innovation. Therefore, we suppose that:

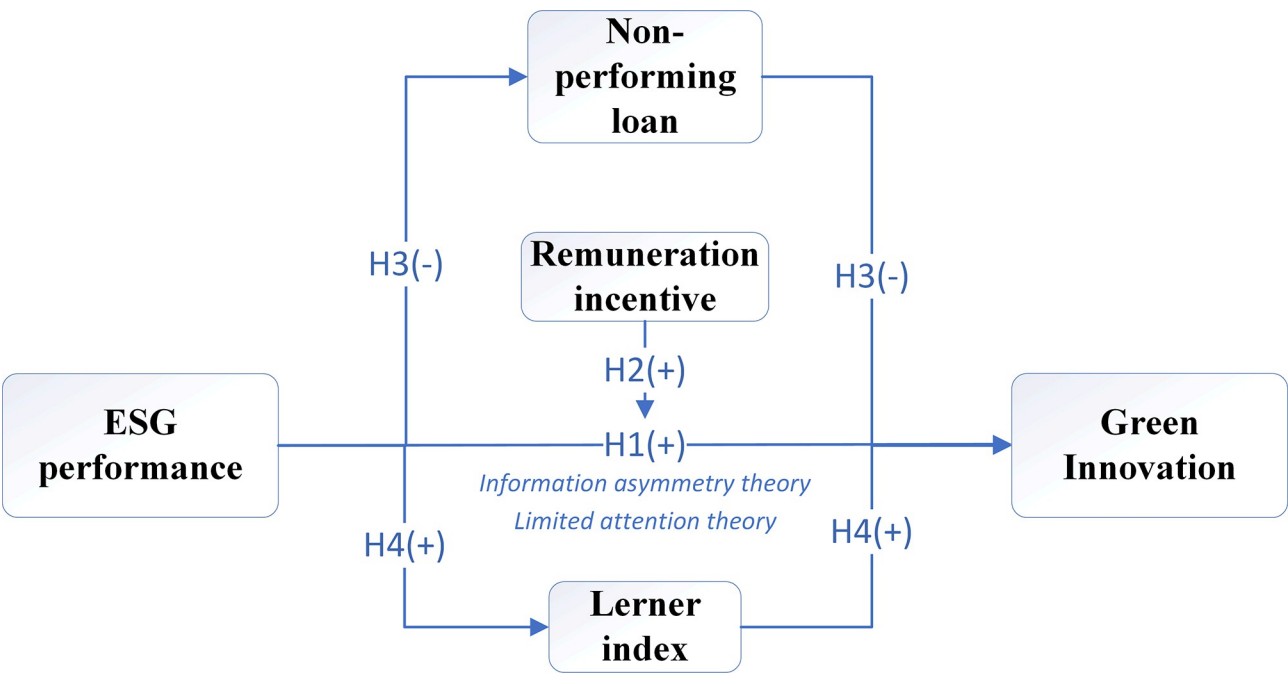

**Fig 1. Theoretical framework diagram.**

**Hypothesis 4.** The Lerner Index plays a mediating role in the process by which commercial banks' ESG performance affects green innovation.

The theoretical framework diagram of this study is shown in Fig 1.

## Data and methodology

### Date

As of 2022, there are a total of 41 bank stocks listed on China's A-share market, specifically including 19 joint-stock commercial banks, 4 policy banks, 4 city commercial banks, 9 agricultural and commercial banks, and 5 village banks. Excluding the banks in which key data are missing, a total of 36 commercial banks obtained valid data. Therefore, this paper selects 2010 to 2021 as the research sample to analyze the impact of ESG performance on green innovation development in China's banking industry. The data used are from the WIND database and the CNRDS database, and some missing data are filled in manually by checking the annual reports of commercial banks.

### Variable definition

**ESG performance (ESG).**   Referring to Yin et al. (2023) [68], we selected the Sino-Securities Index (SNSI) ESG Scores published by ChinDices as the core explanatory variable, and three separate scores for environment (*E*), society (*S*), and governance (*G*) were selected to test the impact of different dimensions on green innovation respectively. Meanwhile, ChinDices publishes ESG Ratings together (9 grades from C to AAA), we assign separate scores to them, and use the results as instrumental variables for endogeneity testing.

**Green Innovation (GRDO).**   The current academic research on enterprise technology innovation measurement indexes focuses on R&D inputs, patent outputs, and R&D personnel ratios [69], which is still applicable in the study of green innovation. Compared with R&D

inputs and R&D personnel ratios, patent outputs are more representative of the degree of development of corporate science and technology innovation [70, 71]. Therefore, this paper selects the annual number of green inventions independently filed (*GRDO*) as a proxy variable for green innovation, while the number is treated as a natural logarithm to ensure the smoothness of the data, which is obtained from the CNRDS database.

**Remuneration Incentives (MW).** Referring to the studies of Conyon and He (2012) [72] and Cui et al. (2021) [73], this paper chooses the compensation received by the company's senior management based on their position and performance, i.e., the annual management wealth (*MW*), as a proxy variable for remuneration incentives; meanwhile, this quantity is treated as a natural logarithm to ensure the smoothness of the data, which is obtained from the WIND database.

**Non-performing Loan (NPL).** Referring to Kryzanowski et al. (2023) [74], the annual non-performing loan ratio (*NPL*), defined as non-performing loans divided by total loans, was selected as one of the mediating variables from the WIND database.

**Lerner index (Lerner).** Referring to Angelini and Cetorelli (2003) [75], the degree of competition of commercial banks is measured using the Lerner index (*Lerner*) based on individual bank data from the WIND database.

**Control variables.** To further improve the explanatory degree of the regression model, drawing on Dietrich and Wanzenried (2011) [76], total bank asset size, gearing ratio, total number of employees, number of years the firm has been in existence, equity checks and balances, and operating growth rate are selected as control variables to be added to the regression, and the above data are obtained from the WIND database. The specific selection of indicators are shown in Table 1.

**Model setting.** We construct the following model:

$$
\begin{aligned}
\mathrm{GRDO}_{it} = {} & \alpha + \beta_1 \mathrm{ESG}_{it} + \beta_2 \mathrm{Size}_{it} + \beta_3 \mathrm{LEV}_{it} + \beta_4 \mathrm{peo}_{it} + \beta_5 \mathrm{ListAge}_{it} + \beta_6 \mathrm{Balance}_{it} \\
& + \beta_7 \mathrm{Growth}_{it} + bank_i + time_t + \varepsilon_{it}
\end{aligned} \tag{1}
$$

Where $\mathrm{GRDO}_{it}$ is the explanatory variable, indicating green innovation; $\mathrm{ESG}_{it}$ is the core explanatory variable, indicating ESG performance; $\mathrm{Size}_{it}$, $\mathrm{LEV}_{it}$, $\mathrm{peo}_{it}$, $\mathrm{ListAge}_{it}$, $\mathrm{Balance}_{it}$, $\mathrm{Growth}_{it}$ is the control variable. We also control for the individual fixed effect $bank_i$ and time fixed effect $time_t$.

**Table 1. Variable definitions.**

| Type | Variable | Measurement | Source of data |
|---|---|---|---|
| Dependent variable | *GRDO* | ln (Annual number of green inventions independently filed+1) | the CNRDS database |
| Independent variable | *ESG* | ESG rating scores | the SNSI ESG Scores published by ChinDices |
| | *E* | Environmental dimension rating scores | |
| | *S* | Social dimension rating scores | |
| | *G* | Governance dimension rating scores | |
| Moderating variable | *MW* | ln (Annual remuneration of management+1) | the WIND database |
| Mediating variable | *NPL* | non-performing loans divided by total loans | |
| | *Lerner* | Lerner index | |
| Control variables | *Size* | Total asset | |
| | *LEV* | Asset-Liability ratio | |
| | *peo* | ln (Total number of employees+1) | |
| | *ListAge* | Years of business establishment | |
| | *Balance* | Shareholding checks and balances | |
| | *Growth* | Revenue growth rate | |

In this model, this study focuses on the coefficient $\beta_1$ of influence of the core explanatory variables $ESG_{it}$ on the explanatory variables $GRDO_{it}$. If the coefficient $\beta_1$ is significantly positive, it represents that the ESG performance of Chinese commercial banks can significantly promote green innovation; on the contrary, if the coefficient $\beta_1$ is significantly negative, it represents that the ESG performance of Chinese commercial banks can significantly inhibit green innovation. According to the previous theoretical analysis, this paper expects the coefficient $\beta_1$ to be significantly positive.

Further, in this paper, in order to explore the moderating effect of management compensation as a moderating variable on the benchmark effect, the model for the mechanism analysis is set up as follows:

$$GRDO_{it} = \alpha + \beta_1 ESG \times MW_{it} + \beta_2 ESG_{it} + \beta_3 MW_{it} + \beta_4 Size_{it} + \beta_5 LEV_{it} + \beta_6 peo_{it}$$
$$+ \beta_7 ListAge_{it} + \beta_8 Balance_{it} + \beta_9 Growth_{it} + bank_i + time_t + \varepsilon_{it} \tag{2}$$

In this model, $MW_{it}$ is the moderator variable, which represents the annual management compensation of commercial banks; $MW \times ESG_{it}$ is the core explanatory variable of the moderating effect, which represents the cross-multiplication of the treatment variable and the moderating variable.

In this model, this study focuses on the influence coefficient $\beta_1$ of $ESG \times MW_{it}$, the cross-multiplier of the treatment variable and the moderator variable, if the coefficient is significantly positive, it represents the positive moderating benchmark effect of the management compensation; on the contrary, if the coefficient is significantly negative, it represents the negative moderating benchmark effect of the management compensation. According to the previous theoretical analysis, this paper expects the coefficient $\beta_1$ is significantly positive.

In addition, in order to explore the role channels of benchmark effect, this paper analyzes two different role channels of non-performing loan ratio and market competitiveness, and the model setting for the analysis of the above channels is as follows:

$$NPL_{it} = \gamma + \gamma_1 ESG_{it} + \gamma_2 Size_{it} + \gamma_3 LEV_{it} + \gamma_4 peo_{it} + \gamma_5 ListAge_{it} + \gamma_6 Balance_{it} + \gamma_7 Growth_{it}$$
$$+ bank_i + time_t + \varepsilon_{it} \tag{3}$$

$$GRDO_{it} = \alpha + \beta_1 NPL_{it} + \beta_2 Size_{it} + \beta_3 LEV_{it} + \beta_4 peo_{it} + \beta_5 ListAge_{it} + \beta_6 Balance_{it}$$
$$+ \beta_7 Growth_{it} + bank_i + time_t + \varepsilon_{it} \tag{4}$$

$$Lerner_{it} = \gamma + \gamma_1 ESG_{it} + \gamma_2 Size_{it} + \gamma_3 LEV_{it} + \gamma_4 peo_{it} + \gamma_5 ListAge_{it} + \gamma_6 Balance_{it}$$
$$+ \gamma_7 Growth_{it} + bank_i + time_t + \varepsilon_{it} \tag{5}$$

$$GRDO_{it} = \alpha + \beta_1 Lerner_{it} + \beta_2 Size_{it} + \beta_3 LEV_{it} + \beta_4 peo_{it} + \beta_5 ListAge_{it} + \beta_6 Balance_{it}$$
$$+ \beta_7 Growth_{it} + bank_i + time_t + \varepsilon_{it} \tag{6}$$

Where $NPL_{it}$ is the mediating variable, which represents the non-performing loan ratio of commercial banks; $Lerner_{it}$ is another mediating variable and represents the Lerner index of commercial banks.

In the mediated effects model of the NPL ratio, this study focuses on the coefficient $\gamma_1$ of the effect of the treatment variable $ESG_{it}$ on the mediating variable $NPL_{it}$, and the coefficient $\beta_1$ of the effect of the mediating variable $NPL_{it}$ on the outcome variable $GRDO_{it}$. Based on the theoretical analysis in the previous section, this paper expects the coefficients $\gamma_1$ to be significantly negative and $\beta_1$ to be significantly negative.

In the mediation effect model of market competition, this study focuses on the coefficient $\gamma_1$ of the treatment variable $ESG_{it}$ on the mediator variable $Lerner_{it}$, and the coefficient $\beta_1$ of

the mediator variable Lerner$_{it}$ on the outcome variable GRDO$_{it}$. Based on the theoretical analysis in the previous section, this paper expects that the coefficients $\gamma_1$ are significantly positive and $\beta_1$ are significantly positive.

## Empirical results and discussions

### Descriptive statistics

This paper provides descriptive statistics for each variable. From the statistical results of 263 observations, the mean value of ESG rating score is 80.129, the minimum value is 65.3, and the maximum value is 87.98, which indicates that the rating index of China's commercial banking industry is basically at the medium to high level. The statistical index characteristics of the remaining data are shown in Table 2.

### The main empirical results

Table 3 shows the regression results of commercial banks' ESG performance on green innovation, in which the regression results are categorized into four groups to verify the coefficients of the impact of the full dimension, environmental dimension, social dimension and governance dimension on green innovation. In column (1), the regression coefficient of ESG overall evaluation score is 0.056, which is significant at 1% level, indicating that ESG performance of commercial banks can significantly contribute to the development of green innovation. In columns (2), (3), and (4), the regression coefficient of the environmental dimension is 0.03, which is significant at the 5% level; the regression coefficient of the social dimension is 0.019, which is significant at the 5% level; and the regression coefficient of the corporate governance dimension is 0.038, which is significant at the 1% level, and it can likewise indicate that the ESG performance of the commercial banks can contribute to the development of green innovation.

This suggests that independent rating agencies construct ESG ratings of commercial banks' social responsibility, which can properly alleviate the problem of internal and external information barriers, and can indeed promote the efficiency of commercial banks' green innovation by improving the information environment. Our empirical findings agree with previous research and theoretical hypothesis one [77, 78].

**Table 2. Descriptive statistics.**

| Variable | N | Mean | SD | Min | Max |
|---|---|---|---|---|---|
| *GRDO* | 263 | 0.92 | 1.458 | 0 | 6.48 |
| *ESG* | 263 | 80.129 | 3.903 | 65.3 | 87.98 |
| *E* | 263 | 70.009 | 5.81 | 56.3 | 88.279 |
| *S* | 263 | 77.847 | 5.616 | 60.59 | 89.49 |
| *G* | 263 | 86.064 | 5.891 | 56.45 | 96.52 |
| *MW* | 263 | 7.616 | 0.601 | 4.796 | 9.053 |
| *NL* | 263 | 1.284 | 0.406 | 0.38 | 2.47 |
| *Lerner* | 263 | 0.356 | 0.157 | 0.002 | 0.683 |
| *Size* | 263 | 10.101 | 1.511 | 6.94 | 12.771 |
| *LEV* | 263 | 92.877 | 1.124 | 89.667 | 96.588 |
| *peo* | 263 | 10.316 | 1.631 | 7.21 | 13.129 |
| *ListAge* | 263 | 1.926 | 0.839 | 0 | 3.434 |
| *Balance2* | 263 | 1.529 | 0.76 | 0.241 | 3.845 |
| *Growth* | 263 | 0.135 | 0.132 | -0.21 | 0.542 |

**Table 3. Baseline results.**

| VARIABLES | GRDO | | | |
|---|---|---|---|---|
| | (1) | (2) | (3) | (4) |
| ESG | 0.056*** | - | - | - |
| | (2.95) | - | - | - |
| E | - | 0.030** | - | - |
| | - | (2.26) | - | - |
| S | - | - | 0.019** | - |
| | - | - | (2.75) | - |
| G | - | - | - | 0.038*** |
| | - | - | - | (2.77) |
| Size | -1.276 | -1.383 | -1.324 | -1.413 |
| | (-1.29) | (-1.38) | (-0.84) | (-1.42) |
| LEV | 0.250** | 0.270** | 0.247 | 0.256** |
| | (2.17) | (2.32) | (1.40) | (2.22) |
| peo | -2.027*** | -1.868*** | -2.015** | -1.935*** |
| | (-3.36) | (-3.08) | (-2.58) | (-3.21) |
| ListAge | -0.695*** | -0.650*** | -0.671** | -0.679*** |
| | (-3.31) | (-3.07) | (-2.13) | (-3.23) |
| Balance2 | -0.549** | -0.574** | -0.610*** | -0.570** |
| | (-2.26) | (-2.34) | (-2.99) | (-2.34) |
| Growth | 0.355 | 0.419 | 0.285 | 0.189 |
| | (0.59) | (0.69) | (0.42) | (0.31) |
| Constant | 5.028 | 5.072 | 8.797 | 6.002 |
| | (0.55) | (0.55) | (0.97) | (0.66) |
| Observations | 263 | 263 | 263 | 263 |
| R-squared | 0.546 | 0.539 | 0.533 | 0.544 |
| Year | Control | Control | Control | Control |
| Bank | Control | Control | Control | Control |

Note: Robust standard errors are reported in parenthesis.

*, **, and *** denote 10%, 5%, and 1% significance levels, respectively.

## Moderating mechanism analysis

Table 4 shows the test of the moderating effect of management compensation. In this regression result, we only need to observe the regression result of the cross-multiplier term between commercial banks' ESG performance and management's annual compensation. The regression coefficient of *ESG×MW* is 4.211, which is significant at the 10% level, indicating that management's compensation enhancement can positively modulate the promotional effect of commercial banks' ESG performance on the development of green innovation.

This suggests that management compensation can indeed be an effective measure to effectively promote ESG performance as a catalyst for green innovation when commercial bank management may be subject to principal-agent risk. This is consistent with the research hypothesis two of this paper and previous studies, for example, Mazouz and Zhao (2019) investigated the causal relationship between executive compensation incentives and firm innovation and found that the stronger the compensation incentives, the more efficient the firm's innovation [79].

**Table 4. Moderating mechanism analysis.**

| VARIABLES | GDRO |
|---|---|
| | (1) |
| ESG×MW | 4.211* |
| | (1.77) |
| ESG | 0.051*** |
| | (2.64) |
| MW | -0.033 |
| | (-0.22) |
| Size | -1.183 |
| | (-1.19) |
| LEV | 0.264** |
| | (2.28) |
| peo | -2.061*** |
| | (-3.43) |
| ListAge | -0.739*** |
| | (-3.48) |
| Balance2 | -0.535** |
| | (-2.19) |
| Growth | 0.186 |
| | (0.31) |
| Constant | 3.954 |
| | (0.43) |
| Observations | 263 |
| R-squared | 0.553 |
| Year | Control |
| Bank | Control |

Note: Robust standard errors are reported in parenthesis.

*, **, and *** denote 10%, 5%, and 1% significance levels, respectively.

## Mediating mechanism analysis

Referring to MacKinnon (2021) [80], Table 5 shows the mediation effect test for NPL ratio and Lerner index. In particular, the (1) and (2) columns show the regression results of commercial banks' ESG performance on NPL ratio and Lerner index; the (3) and (4) columns show the regression results of NPL ratio and Lerner index on green innovation.

On the one hand, it can be seen from the regression results of ESG performance of commercial banks on NPL ratio in the (1) column. The regression coefficient of commercial banks' ESG performance is -0.016, which is significant at the 1% level, indicating that with the improvement of commercial banks' ESG performance can significantly suppress the NPL ratio. Combined with the regression results of NPL ratio on green innovation in the (3) column, it can be seen that the regression coefficient of NPL ratio is -0.666, which is significant at the 5% level, indicating that the NPL ratio can inhibit the development of green innovation in commercial banks. To summarize, commercial banks' ESG performance can enhance the development of green innovation by suppressing the NPL ratio.

This finding suggests that commercial banks with good ESG performance do optimize loan quality and reduce NPL ratios, which in turn simultaneously increases the volume of loans invested in the green sector, which is consistent with hypothesis three of this study and the

Table 5. Mediating mechanism analysis.

| VARIABLES | (1) | (2) | (3) | (4) |
|---|---|---|---|---|
| | NPL | Lerner | GRDO | GRDO |
| ESG | -0.016*** | 0.001* | - | - |
| | (-3.77) | (1.89) | - | - |
| NL | - | - | -0.666** | - |
| | - | - | (-2.27) | - |
| Lerner | - | - | - | 2.503** |
| | - | - | - | (2.70) |
| Size | -0.095 | -0.200** | -1.501 | -0.965 |
| | (-0.42) | (-2.86) | (-1.50) | (-1.54) |
| LEV | -0.060** | 0.002 | 0.222* | 0.259*** |
| | (-2.25) | (0.24) | (1.88) | (3.38) |
| peo | -0.040 | 0.105*** | -1.946*** | -2.165*** |
| | (-0.29) | (2.97) | (-3.21) | (-6.29) |
| ListAge | -0.218*** | 0.037** | -0.812*** | -0.754*** |
| | (-4.52) | (2.29) | (-3.66) | (-3.21) |
| Balance2 | 0.299*** | -0.031** | -0.412 | -0.544*** |
| | (5.35) | (-2.16) | (-1.58) | (-3.59) |
| Growth | -0.554*** | -0.027 | -0.095 | 0.329 |
| | (-4.02) | (-1.10) | (-0.15) | (1.03) |
| Constant | 9.110*** | 1.015 | 14.186 | 6.101 |
| | (4.37) | (1.42) | (1.51) | (1.56) |
| Observations | 263 | 263 | 263 | 263 |
| R-squared | 0.765 | 0.801 | 0.539 | - |
| Year | Control | Control | Control | Control |
| Bank | Control | Control | Control | Control |

Note: Robust standard errors are reported in parenthesis.

*, **, and *** denote 10%, 5%, and 1% significance levels, respectively.

findings of previous studies. For example, Liu et al. (2023) find that banks' good ESG performance improves their loan quality and thus reduces NPL ratios [81], while Pan et al. (2021) verify that social distrust, as represented by NPL ratios, inhibits firms' green innovations, and the findings of this paper's empirical study are in line with those of previous studies [82].

On the other hand, the regression result of commercial banks' ESG performance on Lerner index in the (2) column shows that the regression coefficient of commercial banks' ESG performance is 0.001, which is significant at the 10% level, indicating that commercial banks' ESG performance can have a promoting effect on the improvement of Lerner index. Also in the (4) column, the regression coefficient of Lerner index on green innovation of commercial banks is 2.503, which is significant at the 5% level, indicating that Lerner index has a significant promotion effect on green innovation. To summarize, commercial bank ESG performance has a facilitating effect on green innovation development by enhancing the competitiveness of commercial banks.

This finding suggests that commercial banks' ESG enhances market competitiveness, creates a competitive advantage for banks' green innovations, and promotes green innovation outputs, which is consistent with hypothesis four of this paper's research and previous studies. Tai et al., (2024) found that ESG performance is positively correlated with industry concentration when market demand is convex, and ESG enhances firms' market competitiveness [83],

**Table 6. 2SLS regression: Reverse causality issues.**

| VARIABLES | (1) | (2) |
|---|---|---|
| | first stage | second stage |
| | *ESG* | *GRDO* |
| *ESG* | - | 0.045** |
| | - | (2.13) |
| *ESG_Rating* | 3.970*** | - |
| | (28.20) | - |
| *Size* | 1.197 | -1.314 |
| | (0.72) | (-1.32) |
| *LEV* | -0.028 | 0.253** |
| | (-0.14) | (2.19) |
| *peo* | 0.172 | -2.002*** |
| | (0.17) | (-3.32) |
| *ListAge* | 0.261 | -0.689*** |
| | (0.75) | (-3.28) |
| *Balance2* | -0.929** | -0.563** |
| | (-2.31) | (-2.31) |
| *Growth* | 1.350 | 0.336 |
| | (1.34) | (0.56) |
| *Constant* | 48.477*** | 5.753 |
| | (3.23) | (0.63) |
| **Observations** | 263 | 263 |
| **R-squared** | 0.890 | 0.528 |
| *Year* | Control | Control |
| *Bank* | Control | Control |

Note: Robust standard errors are reported in parenthesis.

*, **, and *** denote 10%, 5%, and 1% significance levels, respectively.

while as early as 1995, Archibugi et al. (1995) found a positive correlation between industry concentration and innovation [84], and Flath (2011) further corroborated this finding [85]. The empirical findings of this paper are also consistent with the above previous research results.

## Reverse causality issues

This paper verifies the reverse causality bias that while ESG affects green innovation, green innovation in commercial banks itself, as a socially responsible behavior, likewise improves banks' ESG performance. Referring to Yin et al. (2023), a two-stage least squares (2SLS) regression was conducted as an instrumental variable after assigning ESG ratings, and Table 6 shows the regression results. From the regression results, in the first stage, the instrumental variables of ESG ratings show a significant facilitating effect on ESG ratings, and in the second stage, the instrumental variables still have a significant facilitating effect on green innovation in commercial banks, proving that the endogeneity test is passed.

## Dynamic panel issues

This paper also uses System Generalized Method of Moments (SYS-GMM) and Difference Generalized Method of Moments (DIF-GMM) to test the dynamic panel bias, i.e., past green

**Table 7. The GMM method: Dynamic panel issues.**

| VARIABLES | GRDO | |
|---|---|---|
| | (1) | (2) |
| | SYS-GMM | DIF-GMM |
| ESG | 0.014*** | 0.026* |
| | (2.68) | (1.96) |
| L.GRDO | 0.572*** | 0.198* |
| | (6.69) | (1.80) |
| Size | -0.069 | -0.186 |
| | (-0.77) | (-1.30) |
| LEV | -0.140*** | -0.224*** |
| | (-2.68) | (-2.75) |
| peo | 0.339*** | 0.546*** |
| | (3.35) | (3.35) |
| ListAge | 0.019 | 0.074 |
| | (0.48) | (1.05) |
| Balance2 | -0.066 | -0.153* |
| | (-1.21) | (-1.90) |
| Growth | -0.350 | -0.325 |
| | (-1.13) | (-1.03) |
| Constant | 9.630** | 15.781** |
| | (2.09) | (2.11) |
| Observations | 224 | 224 |
| AR (1) | 0.018 | 0.041 |
| AR (2) | 0.685 | 0.962 |
| sargan | 0.744 | 0.394 |
| Year | Control | Control |
| Bank | Control | Control |

Note: Robust standard errors are reported in parenthesis.

*, **, and *** denote 10%, 5%, and 1% significance levels, respectively.

innovations may affect future green innovations, and Table 7 shows the regression results. The regression coefficients of commercial banks' ESG performance in the two columns are 0.014 and 0.026, respectively, and are significant at the 1% level and 10% level, respectively, proving that the endogeneity test is passed.

## Replacement of explanatory variables

Finally, in order to further reflect the robustness of the regression results of this study, we also include ESG rating as a proxy variable for the robustness test. As can be seen from the regression results in Table 8, the regression coefficients of ESG rating grade, are all positive and significant at the 5% level, indicating that the proxy variables can still have a promotional effect on the development of green innovation in commercial banks, and the robustness test is passed.

## Conclusion

After studying the influence mechanism of ESG on green innovation of commercial banks, this paper draws the following conclusions:

**Table 8. Replacement of explanatory variables.**

| VARIABLES | GRDO | | |
|---|---|---|---|
| | (1) | (2) | (3) |
| | LSDV | SYS-GMM | DIF-GMM |
| ESG | 0.180** | 0.076** | 0.221** |
| | (2.11) | (1.96) | (2.36) |
| L.GRDO | - | 0.303** | 0.776*** |
| | - | (2.07) | (12.67) |
| Size | -1.260 | -0.161 | 0.265 |
| | (-1.25) | (-1.24) | (1.60) |
| LEV | 0.252** | -0.212*** | -0.175*** |
| | (2.16) | (-2.62) | (-2.65) |
| peo | -1.994*** | 0.481*** | 0.001 |
| | (-3.28) | (2.78) | (0.01) |
| ListAge | -0.677*** | 0.033 | -0.143 |
| | (-3.19) | (0.51) | (-0.74) |
| Balance2 | -0.605** | -0.168** | 0.042 |
| | (-2.48) | (-2.21) | (0.60) |
| Growth | 0.397 | -0.172 | -0.114 |
| | (0.65) | (-0.46) | (-0.30) |
| Constant | 7.952 | 16.794** | 13.057** |
| | (0.87) | (2.27) | (2.20) |
| Observations | 263 | 224 | 224 |
| R-squared | 0.537 | - | - |
| AR (1) | - | 0.074 | 0.024 |
| AR (2) | - | 0.944 | 0.677 |
| sargan | - | 0.817 | 0.816 |
| Year | Control | Control | Control |
| Bank | Control | Control | Control |

Note: Robust standard errors are reported in parenthesis.

*, **, and *** denote 10%, 5%, and 1% significance levels, respectively.

First, ESG has a facilitating effect on the green innovation of commercial banks, and the conclusion is robust to 2SLS and GMM tests for reverse causality and dynamic panel bias. The findings of the study provide useful insights for policymakers and bank management: by enhancing the ESG performance of commercial banks, the development of green innovation can be effectively promoted to realize the goal of sustainable development.

Second, management compensation incentives can positively moderate the causal relationship between ESG performance and green innovation in commercial banks. To further expand this moderating effect, commercial banks can further improve the compensation incentive mechanism for management, incorporate ESG performance into the compensation incentive assessment criteria, and incentivize management to play a positive role in green innovation by giving them incentives related to ESG objectives, thereby promoting the development of green innovation in commercial banks.

Thirdly, this paper investigates the mediating effects of the NPL ratio and Lerner's index in the baseline effect, where commercial banks' ESG performance promotes green innovation by lowering the NPL ratio and increasing the Lerner's index. The results of this study suggest that government departments can promote green innovation by improving the ESG mandatory

disclosure system, nature protection laws, and other laws and regulations, and by improving the transmission efficiency among explanatory, mediating, and interpreted variables.

Based on this, the paper makes the following policy recommendations: First, banks need to set clear ESG objectives and integrate them into their strategic planning and performance assessment systems. This will help to raise management's awareness of ESG issues and ensure that ESG objectives are effectively implemented, thereby promoting green innovation in banks and further addressing the "dual carbon" issue. Second, commercial banks can establish a comprehensive environmental risk management framework, including improving management compensation incentives, environmental due diligence, and sustainable investment. By incorporating environmental risks into management compensation incentives, banks can direct funds to projects and enterprises that are in line with the development of green innovation, thus further promoting incentives for green innovation. Third, commercial banks should actively cooperate and communicate with stakeholders such as the government, regulatory agencies, environmental organizations, and customers, and government departments should actively improve supporting laws and regulations, which can help to promote the transmission efficiency between mediating variables and explanatory variables, and jointly promote the development of green innovation.

## Supporting information

**S1 File. Dataset.** This document is a collection of research data.
(XLSX)

**S2 File. Program commands.** This document is the Stata command.
(DOCX)

## Author Contributions

**Project administration:** Yingchun Zhang, Yang Li.

**Writing – original draft:** Qiliang Wang.

**Writing – review & editing:** Yingchun Zhang, Peihao Wang.

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
