## [Decision Letter · Decision Letter 0]

25 Jun 2024

PONE-D-24-21394ESG performance and green innovation in commercial banksPLOS ONE

Dear Dr. Zhang,

Thank you for submitting your manuscript to PLOS ONE. After careful consideration, we feel that it has merit but does not fully meet PLOS ONE’s publication criteria as it currently stands. Therefore, we invite you to submit a revised version of the manuscript that addresses the points raised during the review process. Based on reviewer's comments I kindly ask you to proceed with minor revision to answer his demands to improve your manuscript.

We look forward to receiving your revised manuscript.

Kind regards,

Marcelo Dionisio

Academic Editor

PLOS ONE

Journal Requirements:

4. We are unable to open your Supporting Information file Data Set.dta and Supplementary Material.do. Please kindly revise as necessary and re-upload.

Reviewers' comments:

Reviewer's Responses to Questions

**Comments to the Author**

1. Is the manuscript technically sound, and do the data support the conclusions?

Reviewer #1: Yes

Reviewer #2: Yes

2. Has the statistical analysis been performed appropriately and rigorously? 

Reviewer #1: Yes

Reviewer #2: Yes

3. Have the authors made all data underlying the findings in their manuscript fully available?

Reviewer #1: Yes

Reviewer #2: Yes

4. Is the manuscript presented in an intelligible fashion and written in standard English?

Reviewer #1: Yes

Reviewer #2: Yes

5. Review Comments to the Author

Reviewer #1: You must discuss the main statistical findings in your study (hypotheses testing result). You need to provide the following points for each hypotheses testing result: a) Result of the study, b) Logical explanation of why the result is positive/ negative/ significant/ insignificant, particularly in your study only, and c) Previous studies supporting your main findings

Reviewer #2: The manuscript appears to be technically sound. The authors conducted an empirical study of 36 Chinese commercial banks over a 12-year period to examine the relationship between ESG performance and green innovation. The data and methodology seem appropriate to support their conclusions about ESG promoting green innovation and the mediating factors involved. The study appears to make a novel contribution by examining ESG performance and green innovation in Chinese commercial banks, including mediating factors. The research question is relevant and timely given the focus on sustainability in the banking sector.

6. PLOS authors have the option to publish the peer review history of their article (what does this mean?). If published, this will include your full peer review and any attached files.

Reviewer #1: No

Reviewer #2: **Yes: **Bayan Mohamad Alshaib

---

## [Author Response · Author response to Decision Letter 0]

1 Jul 2024

Response report

First of all, we would like to express our sincere gratitude to the esteemed editors and the two reviewers for their suggestions and support for this manuscript. These suggestions have further deepened our understanding of this research topic and made the level of the manuscript clearer. We have carefully studied these suggestions and revised the manuscript in the hope that our manuscript will live up to your expectations of us. The revision of the manuscript and the responses to the editors and reviewers are summarized below.

Summary of the revision:

Introduction: We have removed the descriptive text on the origins of ESG and streamlined the "Article Section Settings" section.

Theory and hypotheses: We re-uploaded the research framework diagram (Fig 1) following the editorial formatting requirements.

Data and methodology: We changed the section names, added a data source column to the variable definition table (Table 1), and added numbers to the formulas.

Empirical results and discussions: We have combined the original three chapters on benchmark regression, robustness testing and further research into a single chapter. At the same time, the empirical results of benchmark regression, moderating effect and mediating effect are discussed, and the order of reporting the robustness test is adjusted.

Response to reviewers:

Reviewer #1: 

Introduction: It is too long.

Reply: Thank you very much for the reviewer's suggestion, we re-examined the introduction section carefully, and we found that it was indeed too detailed in the elaboration of ESG data development lineage, therefore, we reorganized the introduction section and deleted the narrative of ESG development lineage, while the literature review section was not deleted in view of the completeness of the literature.After completing the above modifications, the latest introduction logic of this paper is as follows: 1 paragraph introducing the theme, 4 paragraphs of literature review, 1 paragraph of marginal contribution and 1 paragraph of research chapter setting.

Research chapter setting: The steps should be summarized without details 

Reply: Thank you very much for your suggestions and guidance on the chapter setting of this paper. According to your suggestions, we have made the following adjustments and modifications: (1) we have streamlined the text of the chapter settings, eliminated specific details, and retained only the core purpose; (2) we have made some adjustments to the chapters of this paper by deleting the chapter on Robustness testing and further analysis, and combining them into the chapter on Empirical results and discussions, which has made the empirical part of this paper reporting more streamlined.

Data and methodology: (1)The author should insert a column of source of data; (2) Give number for each equation

Reply: We are very grateful to the reviewers for their guidance and suggestions on the variable definition section, for which we have made the following adjustments: (1) added the column of variable source in Table 1 to make the variable definition table clearer; (2) added numbering to the formulas of the article; and (3) re-edited the contents and formatting of the full-text tables, formulas, and pictures to try to make the textual presentation clearer and reduce the difficulty for readers to read.

Empirical results and discussions: (1) You must discuss the main statistical findings in your study (hypotheses testing result). You need to provide the following points for each hypotheses testing result: a) Result of the study, b) Logical explanation of why the result is positive/ negative/ significant/ insignificant, particularly in your study only, and c) Previous studies supporting your main findings; (2) Robustness test should be in the end.

Reply: We are very grateful to the reviewers for their comments and suggestions for changes in the empirical analysis section, and indeed we were lacking in the discussion of the empirical results. In this regard, we have made the following revisions: (1) We have discussed the empirical results for the benchmark regression (H1), moderated effects regression (H2), and mediated effects regression (H3, H4). Part of the theoretical analysis and previous research is added to make the empirical analysis more closely related to the theoretical analysis. (2) The order of reporting the robustness test is adjusted, and the robustness test is reported after the mechanism test, which makes the article clearer.

The original changes to the empirical discussion section are as follows:

H1: This suggests that independent rating agencies construct ESG ratings of commercial banks' social responsibility, which can properly alleviate the problem of internal and external information barriers, and can indeed promote the efficiency of commercial banks' green innovation by improving the information environment. Our empirical findings agree with previous research and theoretical hypothesis one (Ji et al., 2023; Mukhtar et al., 2023).

H2: This suggests that management compensation can indeed be an effective measure to effectively promote ESG performance as a catalyst for green innovation when commercial bank management may be subject to principal-agent risk. This is consistent with the research hypothesis two of this paper and previous studies, for example, Zhao et al., (2023) investigated the causal relationship between executive compensation incentives and firm innovation and found that the stronger the compensation incentives, the more efficient the firm's innovation, which is corroborated by the study of Mazouz and Zhao (2019).

H3: This finding suggests that commercial banks with good ESG performance do optimize loan quality and reduce NPL ratios, which in turn simultaneously increases the volume of loans invested in the green sector, which is consistent with hypothesis three of this study and the findings of previous studies. For example, Liu et al. (2023) find that banks' good ESG performance improves their loan quality and thus reduces NPL ratios, while Pan et al. (2021) and Liu et al. (2023) verify that social distrust, as represented by NPL ratios, inhibits firms' green innovations, and the findings of this paper's empirical study are in line with those of previous studies.

H4: This finding suggests that commercial banks' ESG enhances market competitiveness, creates a competitive advantage for banks' green innovations, and promotes green innovation outputs, which is consistent with hypothesis four of this paper's research and previous studies. Tai et al.,(2024) found that ESG performance is positively correlated with industry concentration when market demand is convex, and ESG enhances firms' market competitiveness, while as early as 1995, Archibugi et al. (1995) found a positive correlation between industry concentration and innovation, and Flath (2011) further corroborated this finding. The empirical findings of this paper are also consistent with the above previous research results.

Reviewer #2: 

The manuscript appears to be technically sound. The authors conducted an empirical study of 36 Chinese commercial banks over a 12-year period to examine the relationship between ESG performance and green innovation. The data and methodology seem appropriate to support their conclusions about ESG promoting green innovation and the mediating factors involved. The study appears to make a novel contribution by examining ESG performance and green innovation in Chinese commercial banks, including mediating factors. The research question is relevant and timely given the focus on sustainability in the banking sector.

Reply: We are very grateful to the reviewers for recognizing this study, and we will make further efforts and perseverance in our future research!

---

## [Decision Letter · Decision Letter 1]

23 Jul 2024

ESG performance and green innovation in commercial banks: Evidence from China

PONE-D-24-21394R1

Dear Dr. Zhang,

We’re pleased to inform you that your manuscript has been judged scientifically suitable for publication and will be formally accepted for publication once it meets all outstanding technical requirements.

Kind regards,

Marcelo Dionisio

Academic Editor

PLOS ONE

Additional Editor Comments (optional):

Reviewers' comments:

Reviewer's Responses to Questions

**Comments to the Author**

1. If the authors have adequately addressed your comments raised in a previous round of review and you feel that this manuscript is now acceptable for publication, you may indicate that here to bypass the “Comments to the Author” section, enter your conflict of interest statement in the “Confidential to Editor” section, and submit your "Accept" recommendation.

Reviewer #1: All comments have been addressed

2. Is the manuscript technically sound, and do the data support the conclusions?

Reviewer #1: Yes

3. Has the statistical analysis been performed appropriately and rigorously? 

Reviewer #1: Yes

4. Have the authors made all data underlying the findings in their manuscript fully available?

Reviewer #1: Yes

5. Is the manuscript presented in an intelligible fashion and written in standard English?

Reviewer #1: Yes

6. Review Comments to the Author

Reviewer #1: The authors have worked on it and updated upon on my request. specially, Empirical results and discussions they have worked on it clearly.

7. PLOS authors have the option to publish the peer review history of their article (what does this mean?). If published, this will include your full peer review and any attached files.

Reviewer #1: No

---

## [Editor Report · Acceptance letter]

2 Aug 2024

PONE-D-24-21394R1 

PLOS ONE

Dear Dr. Zhang, 

I'm pleased to inform you that your manuscript has been deemed suitable for publication in PLOS ONE. Congratulations! Your manuscript is now being handed over to our production team.

Kind regards, 

on behalf of

Professor Marcelo Dionisio 

Academic Editor

PLOS ONE